# New Data on Nephron Microanatomy and Ultrastructure of Senegal Bichir (*Polypterus senegalus*)

**DOI:** 10.3390/biology11101374

**Published:** 2022-09-20

**Authors:** Ekaterina Aleksandrovna Flerova, Evgeniy Georgievich Evdokimov

**Affiliations:** 1Department of Human and Animal Physiology, P.G. Demidov Yaroslavl State University, 150003 Yaroslavl, Russia; 2Federal Williams Research Center of Forage Production & Agroecology, 141055 Lobnya, Russia

**Keywords:** fish, *Polypterus senegalus*, microanatomy, ultrastructure, nephron

## Abstract

**Simple Summary:**

Transitional forms of animals between Pisces and Amphibia are interesting to study, as they are the first to begin the development of a new environment - terrestrial. Such a transitional form are amphibious fish. The study of the structure of the nephron, the main structure that performs an osmoregulatory function, makes it possible to build evolutionary series that describe the processes of transition from the aquatic environment to the terrestrial one. Bichirs is a monophyletic group that arose in the Devonian and formed a species complex only in the Neogene. They share features with lungfish and amphibians, which formed convergently in the early stages of evolution. Therefore Bichirs are of great interest as a transitional form of animals. This study presents new data on the nephron age of *Polypterus senegalus*. Two groups of features are described. The first group consists of ancestral traits that have been preserved in the modern population of *P. senegalus* and are associated with habitat conditions in the aquatic environment. The second group is a complex of characters associated with the adaptation of *P. senegalus* to air breathing and periodic ground migrations.

**Abstract:**

This study presents new data on the microanatomy and ultrastructure of *Polypterus senegalus* nephrons. It was shown that the diameter and ultrastructure of renal corpuscles, a well-developed neck segment, and ultrastructure of two types of epithelial cells of the proximal tubule are ancestral signs of the modern population of *P. senegalus* associated with habitat conditions in the aquatic environment. The outer diameter of the tubules, the height of the epitheliocytes, the presence of two types of epithelial cells of the intermediate and distal tubules of the corresponding ultrastructure, and a large area of nephrogenic tissue are progressive features of the modern population of *P. senegalus*, associated with adaptation to air respiration and periodic terrestrial migrations, which were formed at the early stages of evolution of *P. senegalus* convergently with lungfish and amphibians.

## 1. Introduction

In the comparative evolutionary aspect, it is very important to study the structural and functional features of the nephron, which plays a key role in maintaining homeostasis at the level of osmoregulation. The fish nephron is involved glomerular filtration, reabsorption, and tubular secretion [1]. Nephrons are surrounded by hematopoietic tissue, which to a greater extent performs the function of erythropoiesis [2]. In this aspect, the study of species belonging to the unique group of ray-finned fish, united in the family *Polypteridae*, is very relevant. Bichirs are a monophyletic group that arose in the Devonian and formed a species complex only in the Neogene [3,4,5]. They share characteristics with lungfish that have evolved convergently. *Polypterids* are characterized by the presence of asymmetric dilobed lungs, which are similar in essential features to the lungs of vertebrates, as well as an adapted system for regulating the excretion of metabolites to the conditions of temporary stay in the ground-air environment [6,7]. The use of the body and pectoral fins of *Polypterus senegalus* for movement along terrestrial substrates and the formation of tetrapod-like gait during seasonal migrations from drying temporary spawning lakes to feeding rivers has been proven [8,9].

Living in the ground-air environment became one of the turning points in the evolution of vertebrates, which affected all body systems, including the excretion system [10]. It can be assumed that the nephron of bichirs acquired unique features characteristic of amphibians in the course of evolution. To the best of our knowledge, there have been no comprehensive studies devoted to the study of the microanatomy and ultrastructure of the nephrons of the kidneys of *polypterids*. A promising species, as a model object for solving this problem, is *P. senegalus*, since it can have key features in the structure of the mesonephros, describing the state that occurs during the transition from living in an aquatic environment to terrestrial air.

The purpose of this work is a comprehensive study of the structural organization of the *P. senegalus* Cuvier, 1829 nephron. The realization of this goal is of both theoretical and practical importance. In a theoretical aspect, the data obtained are an important component of comparative evolutionary ideas on the structure of the kidneys of lower vertebrates. In practical terms, the results can be used to understand and improve aquaculture conditions.

## 2. Materials and Methods

### 2.1. Ethical Statement

The care and use of experimental animals complied with the ARRIVE guidelines and was carried out in accordance with EU Directive 2010/63/EU for animal experiments and Russian Federation animal welfare laws. The sampling protocol and procedures employed were ethically reviewed and approved by the Federal Williams Research Center of Forage Production & Agroecology (protocol 2019_10.11).

### 2.2. Fish and Sampling

We used a group of mature diploid individuals of *P. senegalus* Cuvier, 1829, aged 2+.; sex was visually determined. Group were as follows: ten males (17.06 ± 1.62 cm, 14.47 ± 3.12 g) and ten females (13.02 ± 1.29 cm, 13.78 ± 0.61 g). The fish was purchased from a polypterus breeder. The fish was delivered to Russia from the Republic of the Niger. Transportation was carried out in special containers with forced aeration, water temperature 23 °C. The transportation conditions complied with the EU Directive 2010/63/EU for the transportation of animals and the Federal Law of 11 June 2021 N 52-FZ “On the Animal World”.

The fish delivered to the laboratory underwent acclimation in aerated, flow-through stationary aquariums, 200 L in volume, water temperature 23 °C, daily feeding. The care and use of experimental animals complied with the ARRIVE guidelines and was carried out in accordance with EU Directive 2010/63/EU for animal experiments and Russian Federation animal welfare laws. The sampling protocol, procedures employed, and the experiment protocol were ethically reviewed and approved by the P.G. Demidov Yaroslavl State University (protocol number: 19/10, approved on 1 October 2019).

Each individual was injected with the anesthetic benzocaine (supplier Merck Life Science LLC, Darmstadt, Germany). Then, it was measured in fork length (FL) and subjected to neck transection by an experienced person. After that, each animal was dissected and eviscerated.

To study microanatomy, the mesonephros was conditionally divided into 5 segmental parts, the numbering of parts starts from the area of mesonephros located at the gill arches (cranial part) and ends with the area of mesonephros located in the caudal part of the body. In ten individuals (five males and five females), one medial part of the trunk kidney was collected from each fish. In the diagram, this segmental part is numbered 5 (pиc. 1). Samples intended for microanatomy were fixed in 2.5% glutaraldehyde in 0.1 M phosphate buffer, dehydrated in a series of alcohols, purified in xylol and embedded in paraffin according to the standard methodology [11].

Samples intended for histology and ultrastructure analyses were taken out of glutaraldehyde solution and postfixed in 1% osmium tetroxide for 1 h, dehydrated in ethanol and propylene oxide, and embedded in Epon-Araldite (Electron Microscopy Sciences, Hatfield, England) [11]. In ten individuals (five males and five females), one portion of the medial part of the trunk kidney were collected from each fish.

### 2.3. Trunk Kidney Microanatomy

Conventional sections (7 μm) were cut with a SLIDE 2003 microtome (PFM Medical GmbH, Cologne, Germany). Serial sections of the medial part (segmental part 3) were performed in the frontal plane (Figure 1); 108.7 ± 2.21 sections were obtained from each collected kidney sample. A total of 1087 sections were obtained. For the convenience of analysis and comparison of results, the obtained series of sections for each part of the kidney were divided into 6 frontal layers. The numbering of the layers starts from the dorsal to the ventral part of the mesonephros (Figure 1). 

Sections were mounted on glass slides, deparaffinized, and stained with Hematoxylin-eosin [11]. Using a Micromed-6 light microscope (LOMO, Saint Petersburg, Russia) with a Toup Cam 5.1 digital camera, a series of digital photographs was obtained for each section. Morphometric analysis was performed using J Micro Vision 1.2.7 software. On serial sections, a section of the renal corpuscle of the largest diameter was revealed, then, using the standard formula for the surface area of the ball, the area of the outer surface of the renal corpuscles was calculated, assuming that the renal corpuscle is close to the shape of a sphere. On serial sections, the distance between two adjacent capsules was measured, and the number of renal corpuscles on each frontal layer of the mesonephros was taken into account. Density of renal corpuscles was calculated as the ratio of the number of renal corpuscles to the total area of the slice. Three-dimensional reconstruction of the location of the vascular bed and renal corpuscles was performed using Nicolas Roduit, Reconstruct copyright software. To do this, digital photographs of the sections were loaded from the dorsal to the ventral pole of the organ, the necessary structures were isolated, and an image was formed.

### 2.4. Trunk Kidney Histology

Semithin sections (2–3 μm) were cut with a UMTP-3 microtome (Ameqs, Moscow, Russia). Ten sections from each of kidney sample were prepared. Sections were stained with methylene blue. A total of 100 microscopic sections were made; a digital image was obtained for each one. Digital images were taken using a Micromed-6 light microscope (LOMO, Saint Petersburg, Russia) and processed using Image Tool 3.0 software. The outer diameter of the tubules and the diameter of the renal corpuscles were measured. The area of nephrogenic tissues was calculated as the difference between the total area of the section and the sum of the areas of blood vessels and nephron interstitium and expressed as a percentage [11]. From each individual, 20 measurements were obtained for each measured structure.

### 2.5. Trunk Kidney Ultrastructure

Ultrathin sections were cut using a Leica EM UC7 Ultracut (Leica Microsystems, Wetzlar, Germany). Three sections from each portion of the kidney were prepared. A total of 30 sections were created. Sections were stained with uranyl acetate and lead citrate and analyzed using a JEM 1011 transmission electron microscope (JEOL, Tokyo, Japan). A digital image was obtained for each section. Measurements of areas (S) of cells, organelles and inclusions, lengths (l) of the epithelial cells, endocytosis zone and brush border, cilium and microvillus diameters (d) were performed on digital images using Image Tool 3.0 software; from each individual, 10 measurements were obtained for each measured structure. On digital images of cell sections, the number of observed mitochondria, specific granules, vesicles and secretory granules were determined.

### 2.6. Statistical Analysis

To determine the differences between fish groups, statistical analysis was performed in two steps. First, replicates of the data were obtained for each individual; mean values and standard error of the mean were calculated. No statistically significant differences were found between male and female; therefore, males and females were analyzed together at the next step. At this point, mean values and standard error of the mean were calculated for a set of individuals. Statgraphics Plus software was used to analyze data sets. Non-parametric tests were applied, namely, Kruskal–Wallis H-test and Dunn’s multiple comparison post-test. The significance level was set to 0.05 (*p* < 0.05).

## 3. Results

Microanatomy. The renal corpuscles, which form the beginning of the nephron, are mainly located in the area of the kidney adjacent to the spinal column, distal to the main blood vessel, forming a filtration field around them. Most of the tubules are localized at the lateral poles of the mesonephros (Figure 2a–c).

The total number of renal corpuscles with an average diameter of 51.0 ± 0.21 µm in the medial part of the mesonephros is 4699 ± 100. From the ventral to the dorsal pole, the number of renal corpuscles increases. The smallest number of renal corpuscles is observed in the first frontal layer, the largest, in the fifth. The surface area of the renal corpuscles gradually decreases from the ventral to the dorsal pole, in the fourth frontal section this value is minimal, then in the 5–6 frontal layers an increase in the surface area of the renal corpuscles is observed. The largest renal corpuscles were found in the first and second frontal layers. The maximum distance between two adjacent renal corpuscles was found in the second frontal layer, the minimum, in the fourth. It should be noted that the density of renal corpuscles decreases with increasing distance between two adjacent renal corpuscles (Table 1).

The renal corpuscle is followed by the proximal tubule, the beginning of which forms an area consisting of ciliated cells: the neck segment, the average diameter of which is 13.77 ± 0.27 µm. The outer diameter of the proximal tubule increases to 23.0 ± 0.07 µm at the transition to the next section of the tubule, which consists of a cylindrical epithelium, the apical pole of which forms a brush border. The proximal tubule is followed by the distal tubule with an outer diameter of 20.0 ± 0.01 µm. The next increase in outer diameter to 21.93 ± 0.26 µm is observed at the transition of the distal tubule to the connecting tubule (Figure 2d). The proportion of nephrogenic kidney tissues is 77.19 ± 1.29% of the total section area.

**Ultrastructure**. The renal corpuscle consists of a Bowman’s capsule and a glomerulus of capillaries. Bowman’s capsule consists of two sheets: parietal and visceral. The urinary space is 3.77 ± 0.38 µm between the sheets. The capillaries of the glomerulus are of the visceral type (d = 24.33 ± 1.77 μm), the number of capillaries on sections of the glomeruli is 8.23 ± 0.35 (Figure 3a,d). The thickness of the basement membrane of the inner layer of the capsule is 0.09 ± 0.01 μm. Podocytes (S = 36.73 ± 0.36 μm^2^) are located on the basal membrane of the outer surface of the glomerular capillaries; the podocyte body is oval in shape, 9.00 ± 0.14 μm long, with a centrally located nucleus that occupies almost the entire cell section. The nucleus has a rounded shape 19.09 ± 0.27 µm^2^, the heterochromatin is clumpy, and is located to a greater extent along the periphery of the nucleus. A nucleolus was found on sections of the nuclei of some podocytes. The cytoplasm contains 3–5 electron-dense mitochondria with an area of 0.30 ± 0.02 µm^2^, separate cisterns of the rough endoplasmic reticulum, and free ribosomes. The basal part of podocytes forms cytoplasmic processes adjacent to the basement membrane. A filtration gap is formed between the podocyte processes and the membrane (Figure 3b,c,e).

The outer layer of the capsule is built from a single layer of epithelium, located on the basement membrane (Figure 3d). The epithelium of the outer layer of the Bowman’s capsule passes into the epithelium of the proximal tubule.

The wall of the initial section of the proximal tubule is formed by a single layer of ciliated epithelium, forming the neck segment (Figure 3f). Epithelial cells of the neck segment (S = 26.93 ± 1.21 µm^2^) are shaped like a truncated pyramid, 9.45 ± 0.04 µm high. The thickness of the basement membrane in this part of the tubule is 0.13 ± 0.01 µm. The cytoplasm contained a centrally located nucleus (S = 11.62 ± 0.32 µm^2^) with a large number of invaginations (Figure 3h). The brush border is the longest in the neck segment (4.56 ± 0.04 µm) compared to other parts of the proximal tubule and is formed by cilia 0.20 ± 0.00 µm in diameter (Figure 3h). The cytoplasm contains 18–20 oval mitochondria (S = 0.10 ± 0.01 µm^2^). Some of the mitochondria are localized around the nucleus, some of the mitochondria are found in the immediate vicinity of the cilia immersed in the cytoplasm (Figure 3j). The zone of endocytosis is absent. A cell center was found in the apical part of some cells. The apical membrane is even, forming solitary short microvilli (Figure 3i). Intercalated cells were found between the ciliated epithelial cells (Figure 3g). These are oval-shaped cells (S = 35.03 ± 0.26 µm^2^) with an acentrically located nucleus, significantly smaller in size (S = 6.67 ± 0.14 µm^2^) compared to the nucleus of ciliated cells. Individual cisterns of the rough endoplasmic reticulum, large electron-dense lysosomes, and larger mitochondria (S = 0.34 ± 0.01 µm^2^) were found in the cytoplasm. Average number of mitochondria (43.3 ± 0.55) on sections of intercalated cells, is 2 times higher than in the ciliated epitheliocytes (Figure 3g,h).

The next section of the *proximal tubule* is formed by type I epitheliocytes. These elongated, pyramidal-shaped cells are the largest in height—18.99 ± 0.63 µm and area—179.17 ± 9.85 µm^2^ among other proximal tubule cell types (Figure 4a). The basement membrane, on which type I epitheliocytes are located, is wider (0.82 ± 0.02 μm) compared to the basement membrane of the neck segment. Large nuclei (S = 28.97 ± 0.58 µm^2^) of a round shape with single invaginations are located in the basal part of the cell. Heterochromatin is filamentous, evenly distributed along the cut of the nucleus. The cytoplasm contains 22.95 ± 0.49 large (S = 0.61 ± 0.03 µm^2^) rounded mitochondria compared to the neck part, predominantly localized in compartments of the basal labyrinth formed by a smooth endoplasmic reticulum with a cistern width of 0.07± 0.00 µm (Figure 4c). In the apical part of the cells, lysosomes and 18.96 ± 0.5 electron-dense large (S = 0.72 ± 0.03 µm^2^) secretory granules (Figure 4d). At the border with the brush border, there is a 3.40 ± 0.06-µm long zone of endocytosis, characterized by a large number of vesicles and tubules of the smooth endoplasmic reticulum (Figure 4b). The apical pole of the cells carries permanent microvilli 0.11 ± 0.01 µm in diameter, facing the lumen of the tubule and forming a brush border. The brush border is shorter (2.37 ± 0.04 µm) compared to the neck part of the proximal tubule (Figure 4b).

Type II epitheliocytes of the proximal tubule are pyramidal cells, structurally similar to type I cells, equal in base width, but less than those in height 14.75 ± 0.06 µm and area 102.95 ± 4.98 µm^2^ (Figure 4e). The width of the basement membrane is 0.75 ± 0.02 μm and is comparable to the value for type I epitheliocytes. The rounded nucleus (S = 24.78 ± 1.72 µm^2^) is displaced towards the basal part of the cell. Heterochromatin is evenly distributed over the area of the nucleus. The cytoplasm contains 25.37 ± 0.32 larger (S = 1.33 ± 0.1 µm^2^) mitochondria located in compartments of the basal labyrinth compared to type I epithelial cells. In the basal part of the cell, most of the mitochondria are rod-shaped, while closer to the apical pole, the mitochondria are rounded (Figure 4f). The zone of endocytosis is less developed compared to type I cells, its length is 2.83 ± 0.29 µm, although the presence of a tubulo-vesicular system is clearly visible (Figure 4g). The brush border is less high (1.97 ± 0.07 µm) compared to type I cells; the diameter of the microvilli is 0.13 ± 0.00 µm.

When analyzing the cellular structure of the nephron tubule, an intermediate tubule was found at the border between the proximal and distal tubules, the wall of which is formed by a cylindrical epithelium consisting of two types of cells (Figure 5a,d). The basement membrane is the thinnest in this region of the nephron; its width throughout is 0.20 ± 0.03 µm.

Epitheliocytes of type I of the intermediate tubule are smaller, compared to epitheliocytes of the proximal tubule, in height 9.83 ± 0.02 µm and area 63.66 ± 1.23 µm^2^. A significant part of the cytoplasm is occupied by the nucleus (S = 20.43 ± 0.71 µm^2^) displaced towards the basal part of the cell with uneven contours. Heterochromatin is clumpy, condensed largely along the nuclear membrane and in the center of the nucleus (Figure 5a). The cytoplasm contains 27.86 ± 0.74 rounded mitochondria (Figure 5c), the area of which is 0.29 ± 0.2 µm^2^, is more than 2 times smaller than the area of mitochondria of type II epitheliocytes of the proximal tubule. In the cytoplasm, there are electron-transparent vesicles and individual cisterns of the rough endoplasmic reticulum. The apical surface of the cells is formed by short (l = 1.40 ± 0.01 µm) and wide (d = 0.13 ± 0.01 µm) microvilli. There is no formed zone of endocytosis (Figure 5b).

Epitheliocytes of type II of the intermediate tubule are lower (l = 7.65 ± 0.01 μm), with a smaller area, 39.3 ± 0.48 μm^2^, compared with type I epitheliocytes. The nucleus, which is the same as the nuclei of type I epipeliocytes, 20.2 ± 0.52 µm^2^ in area, but with smoother edges, is displaced to the basal part of the cell. The heterochromatin is filamentous, evenly distributed over the nuclear section (Figure 5d). The cytoplasm contains a smaller (14.22 ± 0.2) number of mitochondria compared to type I etipeliocytes, the same size as type I etipeliocytes (0.37 ± 0.01 µm^2^). In the cytoplasm, the largest, in comparison with other parts of the nephron, electron-transparent vesicles and individual cisterns of the rough endoplasmic reticulum were found. The formed zone of endocytosis and the brush border are absent. Separate microvilli, 0.12 ± 0.01 µm in diameter, were found on the apical surface of the cells (Figure 5e).

The wall of the distal tubule is formed by the highest cuboidal epithelium in the nephron, consisting of two types of epithelial cells (Figure 6a,c). The width of the basement membrane throughout this region of the nephron is 0.27 ± 0.07 μm.

Type I epitheliocytes of the distal tubule are the highest; they are 21.24 ± 0.07 µm among the epithelial cells of the nephron cell, with an area of 171.81 ± 2.97 µm^2^ (Figure 6a). Large nuclei (S = 35.7 ± 0.33 µm^2^) are round in shape, displaced towards the basal part of the cells. The cytoplasm of the basal part is characterized by the presence of 74.37 ± 1.08 rod-shaped mitochondria, with an area of 0.22 ± 0.01 µm^2^, located in the compartments of the basal labyrinth formed by the smooth endoplasmic reticulum and oriented along the longitudinal axis of the cell. The zone of endocytosis is absent. The cytoplasm of the apical part of the cell forms lobe-like outgrowths (Figure 6b).

Type II epitheliocytes of the distal tubule are cuboidal cells inferior in height of 12.01 ± 0.19 µm and area of 105.6 ± 1.4 µm^2^ to type I epitheliocytes, despite a wider basal pole (Figure 6c). The nuclei of most cells are displaced to the basal part, they are inferior in area of 21.31 ± 0.67 µm^2^ to the nuclei of type I epitheliocytes. Larger (S = 0.854 ± 0.02 μm^2^) mitochondria in a smaller number (22.75 ± 0.79) compared to type I epitheliocytes are localized similarly to type I epitheliocytes (Figure 6d). This cell type differs from type I epitheliocytes of the distal tubule by the presence in the apical part of a large number of microvilli, 0.27 ± 0.014 µm in diameter, facing the lumen of the tubule (Figure 6e).

The connecting tubule connecting the distal tubule to the collecting duct is formed by flattened cells with even basal and apical plasma membranes (Figure 6g). Some of the cells that form the connecting tubule are cells with a more electron-transparent cytoplasm (Figure 6h), some with an electron-dense, heterogeneous cytoplasm (Figure 6f). Both types of cells have the same height, 6.64 ± 0.05 µm, and area 41.04 ± 0.44 µm2. The height and area of cells is the smallest among the considered types of epitheliocytes. These cells are characterized by a large (S = 23.09 ± 0.26µm^2^), centrally located nucleus and a small number of organelles. In the perinuclear zone, tubules of the rough endoplasmic reticulum, a small number (14.05 ± 0.14) of mitochondria, with an area of 0.10 ± 0.01 µm^2^, were found. The basal labyrinth is absent, the apical surface is almost even. The apical pole is covered with short and irregular microvilli (Figure 6f,h).

## 4. Discussion

Histological analysis has shown that the bichir renal corpuscle has the same structure as both lungfish and freshwater teleosts and amphibians [12,13,14,15]. It has been established that the cut diameter of the renal corpuscle increases in the series *P. senegalus*—freshwater bony fish *Ctenopharyngodon idella*; *Channa punctatus*—amphibians *Rana cancrivora; Triturus (Cynops) pyrrhogaster* [13,15,16]. 

It can be assumed that the filtration activity of the kidney of *P. senegalus* is comparable to that of freshwater teleosts and inferior to that of amphibians. Based on the morphometric data of the renal corpuscles, as well as their spatial arrangement, it can be assumed that in the medial part of the mesonephros of *P. senegalus*, the filtration activity is the highest in the capsules localized in the layers of the dorsal pole.

It should be noted that the outer diameter of the tubules increases in the series *P. senegalus*—freshwater bony fish *C. idella*, *C. punctatus*—lungfish Protopterus dolloi—amphibians R. cancrivora, T. (Cynops) pyrrhogaster [12,13,15,16]. It is known that the outer diameter of the tubules depends both on the lumen of the tubules and on the height of the epitheliocytes that form the tubules. The height of nephron epithelial cells increases in the following order: *P. senegalus*—freshwater bony fish—*C. punctatus*, Coregonus migratorius—lungfish *P. dolloi*—amphibian Bufo bufo [15,17,18,19]. 

The development of nephrogenic tissues in P. senegalus is twice as high as in freshwater teleosts [20]. Amphibians, dissimilar to fish, lose the renal-splenic type of hematopoiesis due to the onset of the transition of the hematopoietic function to a new structure, red bone marrow, so the mesonephros is formed by nephrogenic tissues [21]. Probably, the increase in nephrogenic tissues in the mesonephros of *P. senegalus*, in comparison with freshwater teleosts, is associated with an increase in the total filtration and reabsorption area of the nephron structural units. The morphological basis of this functional feature can probably be both a greater number of nephrons in the *P. senegalus* kidney and a greater length of nephron tubules. In the process of phylogenetic development, a similar structural feature is observed in amphibians, the mesonephros of which has become more specialized. Thus, a large area of nephrogenic tissue is a progressive feature of the modern *P. senegalus* population associated with adaptation to air respiration and periodic terrestrial migrations, which were formed in the early stages of *P. senegalus* evolution convergently with lungfish and amphibians.

A well-developed neck segment has been described already in the most ancient group of vertebrates, the lamprey *Entosphenus japonicas* [22]. It was found in the kidneys of the lungfish *P. dolloi*, sturgeons, and *Esox lucius* [19,23,24,25]. It should be noted that the ultrastructure of the neck segment is similar to the structure of this part of the nephron in fish in the mesonephric kidneys of tailless and caudate amphibians (*Rana temporaria L.*, *Ambystoma maculatum*), leading a conditionally terrestrial lifestyle [26,27]. In the metanephros of mammals, this segment is strongly reduced or absent [10,14,28,29]. The ciliated epithelium of the neck segment of the anamnium is involved in the active movement of the filtrate inside the tubule under conditions of low transmembrane current during filtration in the glomerulus [30]. Under conditions of life in the air-terrestrial environment in the renal corpuscle of the nephron, the strength of arterial pressure, transmembrane current and the rate of formation of primary urine increase, which probably leads to a reduction in the neck region [31]. Thus, a well-developed neck segment of the nephron is an ancestral feature of the modern population of *P. senegalus* associated with living conditions in the aquatic environment.

The proximal tubules are an essential part of the nephron in all groups of cyclostomes, fish, and amphibians. In bichir, as well as in the lungfish *P. dolloi*, freshwater bony fish, and amphibians, they are represented by two types of cells with a common structural plan [13,14,15,19,21,26,32]. Thus, the presence of a developed mechanism of reabsorption in the proximal tubule of the nephron was formed at the early stages of the evolution of lower vertebrates, which affected the conservatism of this part of the nephron, regardless of the environmental conditions of fish and amphibians.

The intermediate tubule has been found already in the lamprey Lampetra fluviati-lis, and also of the more evolutionarily young freshwater fish Abramis ballerus, Stizostedion volgense, the tubule of which is formed by cells flattened to 8 µm, with a smooth apical surface and very sparse and short single microvilli [33]. In the lungfish *P. dolloi*, the intermediate segment is formed by one type of cuboidal villous epithelium cells with a well-developed brush border. The height of the villous cells of *P. dolloi* is 1.5 times that of the villous cells of *P. senegalus* [23]. Using the example of *Geotrypetes seraphini, Notophthalmus viridescens*, *A. maculatum*, it was shown that in amphibians the intermediate segment is represented by low (7–8 µm) cubic ciliated cells [21,27,32]. Our study showed that the ultrastructure of type I cells of the intermediate tubule has the same structure as epitheliocytes of *P. dolloi*. Moreover, the intermediate tubule of *P. senegalus* is more differentiated than that of *P. dolloi*, as evidenced by the presence of two types of epitheliocytes with different degrees of development of the villous epithelium. It is known that the development of brush-bordered nephrons is a consequence of weak transmembrane pressure in the glomerus in lower vertebrates. Some authors hypothesize that a weak fluid flow in the tubules leads to the need for an additional segment to increase the rate of urine transport in the nephrons. In addition, flattening of the cells of the intermediate tubule during evolution makes it possible to shorten the path that water passes through the epithelial cell wall [30,33]. The differentiation of the intermediate tubule of *P. senegalus*, which is more characteristic of higher vertebrates, is a progressive sign of adaptation to changing environmental conditions [33].

The distal tubule is an orthodox part of the nephron pattern and has been noted in all vertebrates [14]. Previous studies have shown that the complexity of the distal tubule may be different for different systematic groups. Thus, for freshwater fish teleost *C. idella*, only one type of cells of the distal tubule was noted, with a characteristic high density of mitochondria and a reduced ciliated apparatus [15]. At the same time, in the lungfish *P. dolloi* and the amphibian *R. cancrivora*, two types of cells were found in the distal tubule of the nephron. The main differences reported by the authors between these cell types were in different cell shapes and sizes [13,19]. The number of types of epitheliocytes in the distal tubule of *P. senegalus*, as well as their ultrastructure, are similar to those of lungfish and amphibians. The high specialization of the distal tubule is also an evolutionary advantage, increasing the eurybiont nature of the species.

The connecting tubule connects the distal tubule to the collecting duct, the embryonic origin of this segment is not considered in our article. The ultrastructure of the *P. senegalus* epitheliocytes forming this region is similar to that of the collecting ducts of the freshwater fish *C. idella*, the amphibious fish *P. dolloi*, and the amphibians *R. cancrivora, Rana temporaria* [13,15,19,33].

## 5. Conclusions

Comparative analysis of the structural organization of the nephron of *P. senegalus* with several other freshwater teleosts, lungfish, and amphibians showed that the diameter and ultrastructure of renal capsules, as their spatial arrangement, a well-developed neck segment, ultrastructure of two types of epithelial cells of the proximal tubule are ancestral features of the modern Senegal bichir population associated. These features are associated with the filtration activity of the kidney in the conditions of life fishes in the freshwater environment.

The outer diameter of the tubules, the height of the epitheliocytes, the presence of two types of epithelial cells of the intermediate and distal tubules of the corresponding ultrastructure, a large area of nephrogenic tissue is a progressive feature of the modern population of *P. senegalus*. These features are associated with adaptation to air respiration and periodic terrestrial migrations, which were formed at the early stages of evolution of *P. senegalus* convergently with lungfish and amphibians.

## Figures and Tables

**Figure 1 biology-11-01374-f001:**
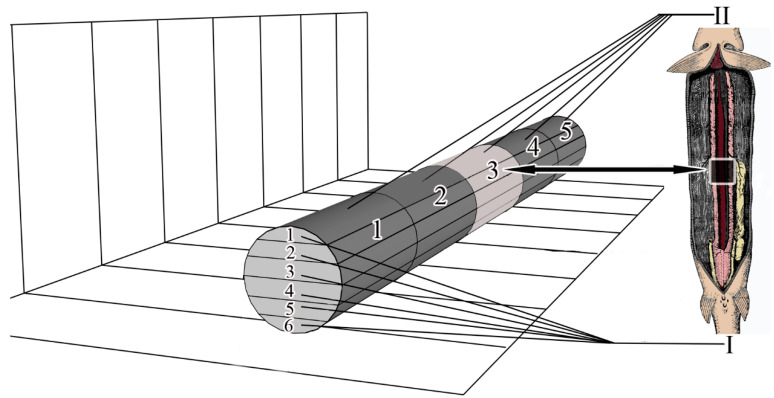
Scheme of axes and planes of the mesonephros. I—layers (1–6), II—segmental parts (1–5).

**Figure 2 biology-11-01374-f002:**
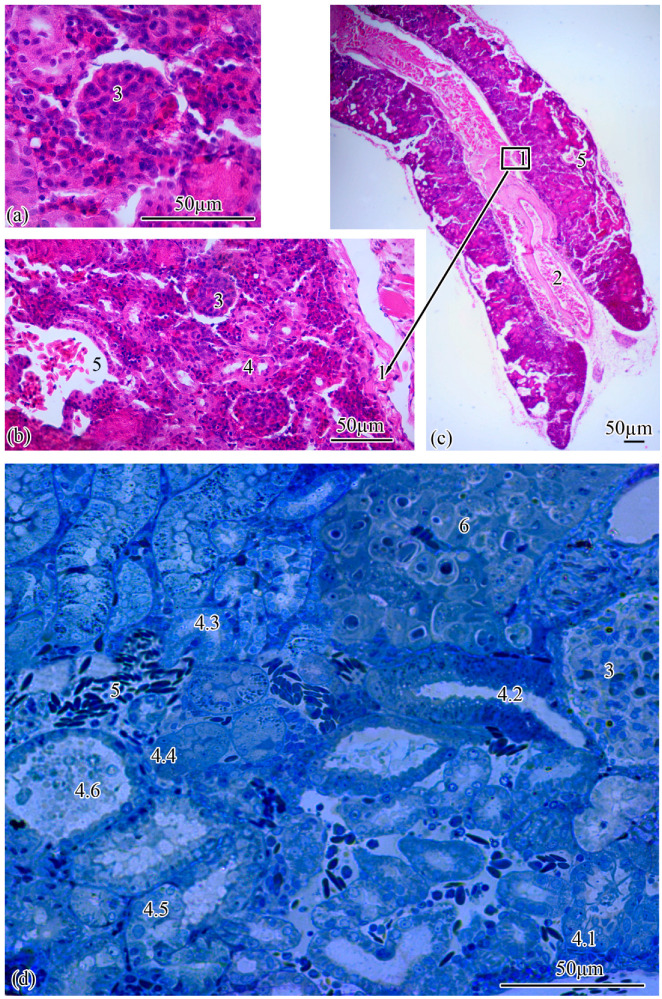
Microanatomy of the trunk kidney (**a**–**d**). Main structural units: wall of the magistral blood vessel (1), magistral blood vessel (2), renal corpuscle (3), nephron tubules (4), neck segment (4.1), part of the proximal tubule formed by type I epithelial cells (4.2), part of the proximal tubule formed by type II epithelial cells (4.3), part of the distal tubule formed by type I epithelial cells (4.4), part of the distal tubule formed by type II epithelial cells (4.5), collecting tubule (4.6), peripheral blood vessels (5), renal interstitium (6).

**Figure 3 biology-11-01374-f003:**
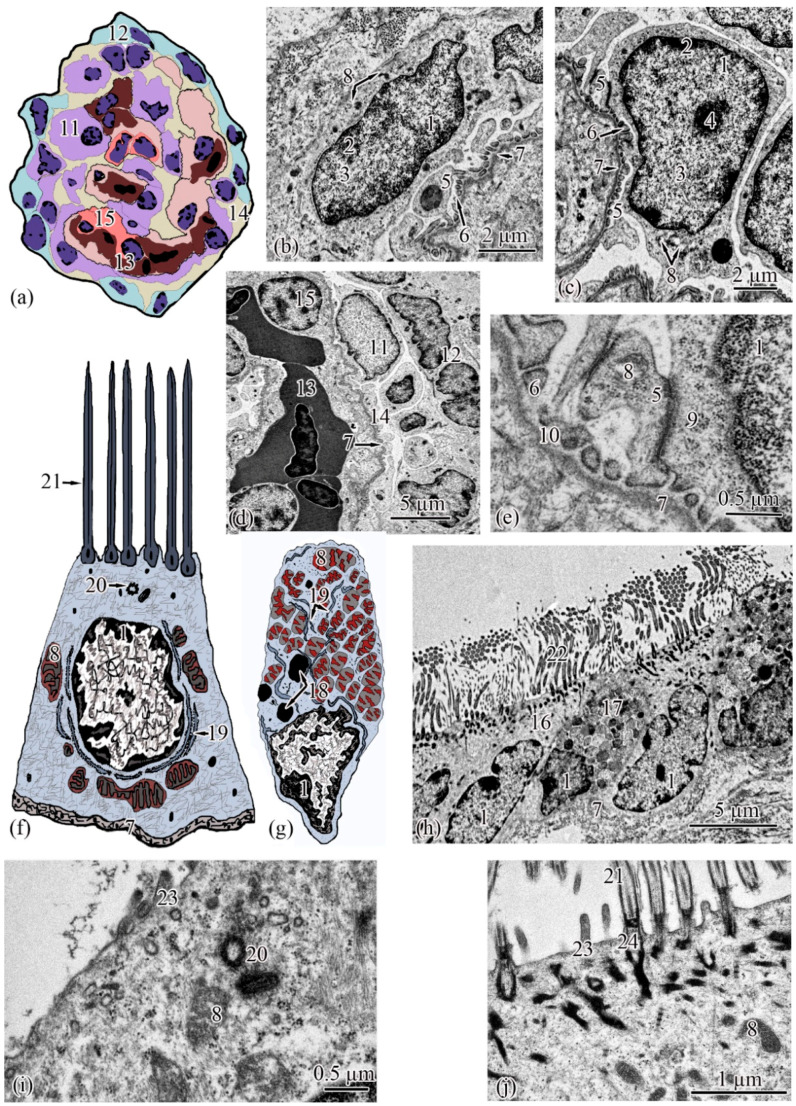
Ultrastructure of the renal corpuscle end of the neck segment. Renal corpuscle scheme (**a**); squamous epithelial cell of the parietal layer of the Bowman’s capsule (**b**); podocyte (**c**); part of the renal corpuscle (**d**); part of the podocyte (**e**); ciliated epithelial cell scheme (**f**); intercalated epithelial cell scheme (**g**); part of the neck segment (**h**); basal part of ciliated epithelial cell (**i**,**j**). Cell units: nucleus (1), heterochromatin (2), euchromatin (3), nucleolus (4), cytotrabecula (5), cytopodia (6), basement membrane (7), mitochondria (8), ribosomes (9), filtration slit (10), podocyte (11), squamous epithelial cell (12), glomerular capillary (13), urinary space (14), endothelial cell (15), ciliated epithelial cell (16), intercalated epithelial cell (17), secretory granules (18), smooth endoplasmic reticulum (19), centrioles (20), cilia (21), brush border (22), microvilli (23), basal body of the cilium (24).

**Figure 4 biology-11-01374-f004:**
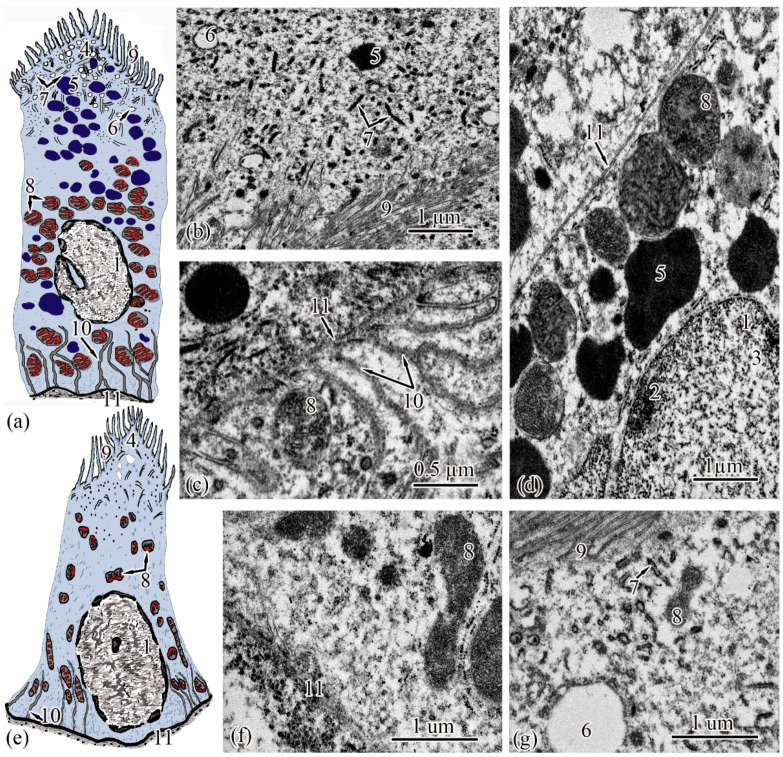
Ultrastructure of the proximal tubule. Type I epithelial cell scheme (**a**); endocytosis zone and a portion of the brush border of type I epithelial cell of the proximal tubule (**b**); basal part of type I epithelial cell of the proximal tubule (**c**); apical part of type I epithelial cell of the proximal tubule (**d**); type II epithelial cell scheme (**e**); basal part of type II epithelial cell of the proximal tubule (**f**); apical part of type II epithelial cell of the proximal tubule (**g**). Cell units: nucleus (1), heterochromatin (2), euchromatin (3), endocytosis zone (4), secretory granules (5), vesicle (6), tubular endoplasmic reticulum (7), mitochondria (8), microvilli (9), smooth endoplasmic reticulum (10), basement membrane (11).

**Figure 5 biology-11-01374-f005:**
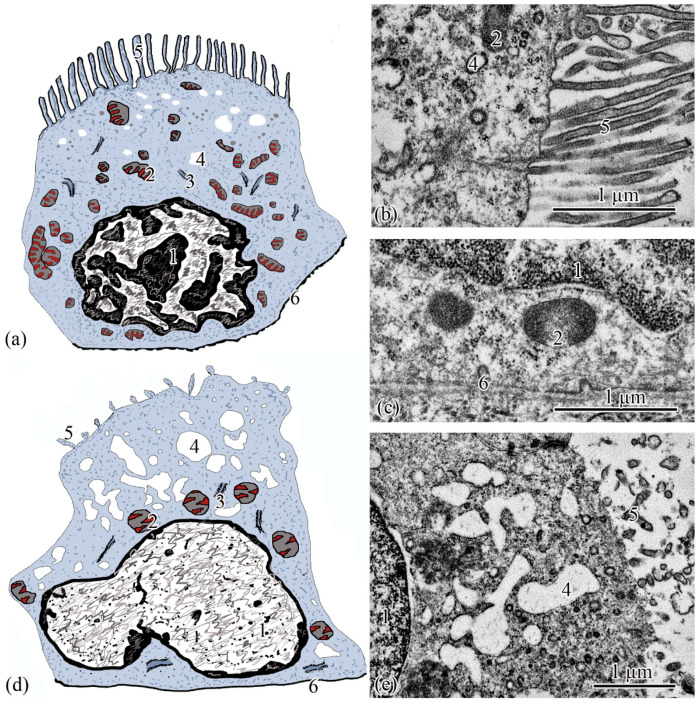
Ultrastructure of the intermediate tubule. Type I epithelial cell scheme (**a**); apical part of type I epithelial cell (**b**); basal part of type I epithelial cell (**c**); type II epithelial cell scheme (**d**); apical part of type II epithelial cell (**e**). Cell units: nucleus (1), mitochondria (2), cisterns of the rough endoplasmic reticulum (3), vesicle (4), microvilli (5), basement membrane (6).

**Figure 6 biology-11-01374-f006:**
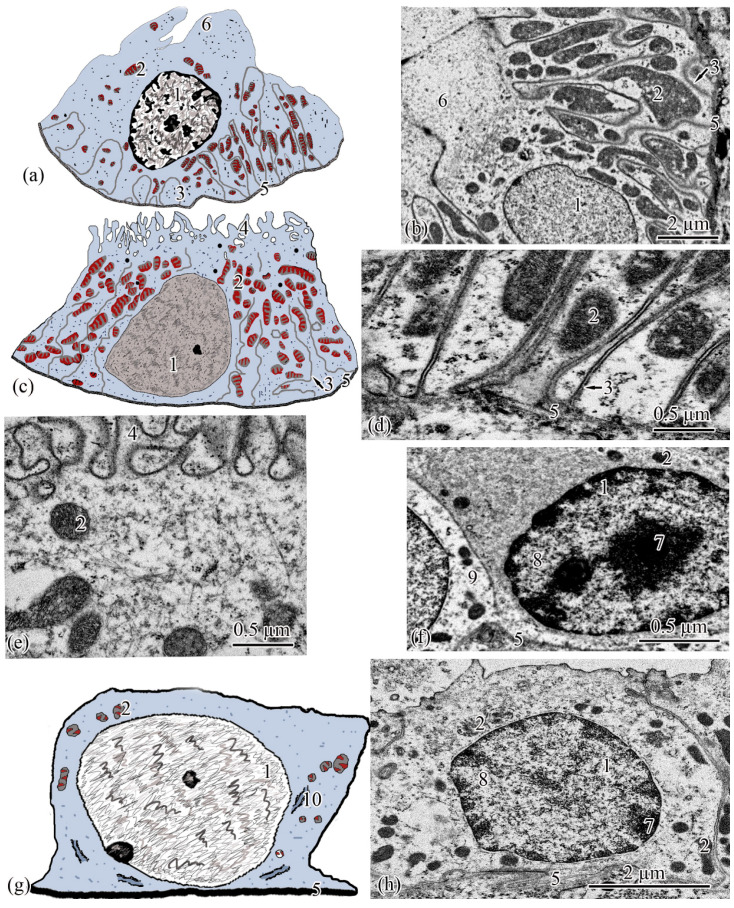
Ultrastructure of the distal tubule and of the connecting tubule. Type I epithelial cell of the distal tubule scheme (**a**); type I epithelial cell of the distal tubule (**b**); type II epithelial cell of the distal tubule scheme (**c**); basal part of type II epithelial cell of the distal tubule (**d**); apical part of type II epithelial cell of the distal tubule (**e**); dark epithelial cell of the connecting tubule (**f**); epithelial cell of the connecting tubule scheme (**g**); light epithelial cell of the connecting tubule (**h**). Cell units: nucleus (1), mitochondria (2), smooth endoplasmic reticulum (3), microvilli (4), basement membrane (5), lobed cytoplasmic process (6), heterochromatin (7), euchromatin (8), part of dark epithelial cell of the connecting tubule (9), cisterns of the rough endoplasmic reticulum (10).

**Table 1 biology-11-01374-t001:** Morphometric parameters of the structural elements of the mesonephros of *P. senegalus*.

Laywer	Capsules Quantity	Square Capsuls, µm^2^	Distance between Capsuls, µm	Density of Renal Corpuscles, Capsules Quantity/ 1 µm^2^ of Tissue
1	466.76 ± 111.55 ^abc^	9892.36 ± 123.39 ^a^	246.32 ± 11.51 ^a^	0.13 × 10^−3^ ± 0.18 × 10^−3 a^
2	611.22 ± 129.3 ^abc^	9576.77 ± 124.55 ^a^	304.72 ± 9.78 ^a^	0.06 × 10^−3^ ± 0.02 × 10^−3 abf^
3	800.95 ± 177.72 ^ab^	8696.91 ± 135.28 ^a^	221.8 ± 6.73 ^a^	0.11 × 10^−3^ ± 0.07 × 10^−3 c^
4	825.22 ± 183.0 ^ac^	6723.63 ± 110.54 ^a^	213.99 ± 5.69 ^a^	0.19 × 10^−3^ ± 0.13 × 10^−3 acdef^
5	1097.92 ± 257.14 ^abc^	8382.83 ± 95.32 ^a^	157.97 ± 5.17 ^a^	0.09 × 10^−3^ ± 0.03 × 10^−3 d^
6	897.0 ± 198.08 ^abc^	8389.25 ± 47.7 ^a^	250.19 ± 7.38 ^a^	0.13 × 10^−3^ ± 0.09 × 10^−3 be^

Note: Values with the same letter indices are significantly different from each other, *p* ≤ 0.05.

## Data Availability

The datasets used in this study are publicly available at https://disk.yandex.ru/d/LpxOY1Layp8dUg, accessed on 18 July 2022.

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
