# Peer review of "New Data on Nephron Microanatomy and Ultrastructure of Senegal Bichir (Polypterus senegalus)"

_biology, 2022, doi:10.3390/biology11101374_

Round 1

Reviewer 1 Report

This study it is an intensive ultrastructural study that describe the organisation of the nephron along the mesonephros in the bichir. The study can be used as a reference to compare with different species, although Senegal bichir is an unusual subject in research.

The study could be beneficiated if the authors consider including and additional figure with a picture or an outline of the bichir and the location of the kidney along the shape of the animal and showing what part correspond to the mesonephros that has been studied.  

11: their ultrastructure are ancestral signs of the modern population… rewrite

49: Keeping and breeding. Not a common way to express in aquaculture setting.  May be… to understand and improve aquaculture conditions/zootechnical handling.  

59: we use a sample of mature. The word sample here is confusing, may be batch, group or something similar will be more clear  

60: samples were as follow: rewrite. The samples ranged between 17.6-14.47… and always use the small size first. The same with the males.

61: the fish was cheered up… rewrite.

74: the destruction of the brain… disturbing description. Rewrite

76: after that an individual was… It must be better “each animal/bichir/individual

78: the organ… change to “the kidney”

83: it is unusual to fix for paraffin sections with glutaraldehyde because it does not complete fix lipids. It is preferable formalin-based fixatives, but it is only a commentary.   

93: semithin sections. Semithin sections ranged from 0,5 to 2 microns and are aimed for location before ultrathin sections. 7 um is a conventional section, not especially thin.

Figure 1. maybe only dividing the mesonephros in layers from ventral to dorsal (II) and the lateral divisions (V) it will be easier to understand. The others are conventional planes not compulsory. In addition, the dorsal and ventral pole seems to be wrong.

116 (2-3 um) here is adequate

126: three sections from each part of the kidney. If there is 5 parts it means, 15 sections from ten individuals. ¿30 sections include male and female, it is right?

337 and 388: doble check if “plan” is the best word here

435-440: too long sentence. Rewrite

441-446: Too long sentence. Rewrite

Author Response

Dear Editor.

On behalf of all authors, I would like to resubmit a manuscript titled “New data on nephron microanatomy and ultrastructure of Senegal bichir (Polypterus senegalus)” by Ekaterina Flerova, Evgeniy Evdokimov, for consideration to be published as an Article in the Biology.  We carefully considered all the suggestions provided by the reviewers and made changes to the text. Please consider the reviewer’s comments and our replies.

Reviewer #1

Comment 1: 11: their ultrastructure are ancestral signs of the modern population… rewrite

Reply: It was shown that the diameter and ultrastructure of renal corpuscles, a well-developed neck segment, ultrastructure of two types of epithelial cells of the proximal tubule are ancestral signs of the modern population of P. senegalus associated with habitat conditions in the aquatic environment.

Comment 2: 49: Keeping and breeding. Not a common way to express in aquaculture setting.  May be… to understand and improve aquaculture conditions/zootechnical handling.  

Reply: In practical terms, the results can be used to understand and improve aquaculture conditions

Comment 3: 59: we use a sample of mature. The word sample here is confusing, may be batch, group or something similar will be more clear

Reply: 65-66 the sample was replaced by a group

Comment 4: 60: samples were as follow: rewrite. The samples ranged between 17.6-14.47… and always use the small size first. The same with the males.

Reply: These indicators indicate the average value and standard error of the mass and length of the body of each individual in the sex group.

Comment 5: 61: the fish was cheered up… rewrite.

Reply: 68: The fish was purchased from a polypterus breeder.

Comment 6: 74: the destruction of the brain… disturbing description. Rewrite

Reply: 80: The each individual was injected with the anesthetic benzocaine (supplier Merck Life Science LLC, Germany).

Comment 7: 76: after that an individual was… It must be better “each animal/bichir/individual

Reply: 82 After that, each animal was dissected and eviscerated.

Comment 8: 78: the organ… change to “the kidney”

Reply: 84 To study microanatomy, the mesonephros…..

Comment 9: 83: it is unusual to fix for paraffin sections with glutaraldehyde because it does not complete fix lipids. It is preferable formalin-based fixatives, but it is only a commentary.

Reply: We express our gratitude to the respected reviewer, we will definitely take this fact into account in future works.

Comment 10: 93: semithin sections. Semithin sections ranged from 0,5 to 2 microns and are aimed for location before ultrathin sections. 7 um is a conventional section, not especially thin.

Reply: 93 Conventional sections (7 μm) were cut with a SLIDE 2003 microtome (Germany). Serial sections of the medial part (segmental part 3) were performed in the frontal plane (Fig. 1).

Comment 11: Figure 1. maybe only dividing the mesonephros in layers from ventral to dorsal (II) and the lateral divisions (V) it will be easier to understand. The others are conventional planes not compulsory. In addition, the dorsal and ventral pole seems to be wrong.

Reply: The figure and its captions have been changed in accordance with the recommendations of the reviewer

Comment 12: 126: three sections from each part of the kidney. If there is 5 parts it means, 15 sections from ten individuals. ¿30 sections include male and female, it is right?

Reply: From each kidney obtained, only from the medial section, three sections were excised and taken for analysis. A total of 5 males and 5 females were used. Therefore, there are 30 parts in total. 3 * (5 + 5) = 30

Comment 13: 337 and 388: doble check if “plan” is the best word here

Reply: 350, 351 Histological analysis has shown that the bichir renal corpuscle has the same structural common with both lungfish and freshwater teleosts and amphibians 435-440: too long sentence. Rewrite

Comment 14: 441-446: Too long sentence. Rewrite

Reply: 421-423 The offer has been shortened

Reviewer 2 Report

The manuscript entitled “New data on nephron microanatomy and ultrastructure of Senegal bichir (Polypterus senegalus)” is an important study because the comparative evolutionary aspect, it is very important to study the structural and functional features of the nephron, which plays a key role in maintaining homeostasis at the level of osmoregulation. The fish nephron is involved in both incretory and excretory activities. This multifaceted activity is provided by glomerular filtration, reabsorption, and tubular secretion. In this aspect, the study of species belonging to the unique group of ray-finned fish, united in the family Polypteridae, is very relevant.

Write this part clearly, please, it is relevant to your research work ... there are words repaated to the others, also why are you doing these quotes ??? !!!

The abstract is relevant but please write clearly.

The introduction is relevant but must include new references and it is recommended to rewrite this part clearly so that the results can be clearly expressed. 

The discussion, in the light of results and knowledge, is relevant. 

Your manuscript will not be accepted unless both the technical and grammatical revisions have been made successfully.

Based on these comments, I recommend a moderate revision of analytical aspects of this manuscript before final decision about its acceptance.

Minor comment:

1.   Introduction:

Rewrite this part with new references it is a very old research (e.g., Near T.J., et al., 2012) and (Offem B. O., et al., 2011),

Please consider if it would be useful to indicate other analyzes besides those published …see and insert as below:

See reference: Francesco Fazio, Concetta Saoca, Gregorio Costa, Alessandro Zumbo, Giuseppe Piccione, Vincenzo Parrino (2019) - Flow cytometry and automatic blood cell analysis in striped bass Morone saxatilis (Walbaum, 1792): A new hematological approach, Aquaculture 513 – 734398. 

2.   Materials and Methods:

Fish and sampling.

Please consider whether it would be useful to indicate other chemical analyzes such as the indication of toxic substances and concentrations ... The fish was delivered to Russia from the Republic of Niger. Transportation was carried out in special containers with forced aeration, water temperature 23 ° С. !!… See (Table 1) was the experiment protocol also developed by you with due care? Double-check the steps.

Statistical:

The statistical analysis used is appropriate.

3.   Results:

The authors should be reduce this part, please the results shown in figures (2a -c ).

4.   Discussion:

This part is very long, reduce it and discuss only the obtained results.

Unfortunately, due to the lack of publications, we were unable to build similar series for such indicators as the number of capsules and their surface area. However, based only on the data on section diameters, it can be assumed that the filtration activity of the kidney of P. senegalus is comparable to that of freshwater teleosts and inferior to that of amphibians. ....

... there are other researches that assure this result ...

Our study showed that the ultrastructure of type I cells of the intermediate tubule 406 has a common structural plan with the epitheliocytes of P. dolloi.

5. Conclusion:

It would also be useful to include in the conclusions all the samples analyzed that showed different section diameters in order to assume that the filtration activity of the P. senegalus kidney is similar to other fish…Explain this step better.

Author Response

Dear Editor.

On behalf of all authors, I would like to resubmit a manuscript titled “New data on nephron microanatomy and ultrastructure of Senegal bichir (Polypterus senegalus)” by Ekaterina Flerova, Evgeniy Evdokimov, for consideration to be published as an Article in the Biology.  We carefully considered all the suggestions provided by the reviewers and made changes to the text. Please consider the reviewer’s comments and our replies.

Reviewer #2

Comment 1: 1. Introduction:

Rewrite this part with new references it is a very old research (e.g., Near T.J., et al., 2012) and (Offem B. O., et al., 2011),

Please consider if it would be useful to indicate other analyzes besides those published …see and insert as below:

Reply: See reference: Francesco Fazio, Concetta Saoca, Gregorio Costa, Alessandro Zumbo, Giuseppe Piccione, Vincenzo Parrino (2019) - Flow cytometry and automatic blood cell analysis in striped bass Morone saxatilis (Walbaum, 1792): A new hematological approach, Aquaculture 513 – 734398. 

Comment 2: 2. Materials and Methods:

Fish and sampling.

Please consider whether it would be useful to indicate other chemical analyzes such as the indication of toxic substances and concentrations ... The fish was delivered to Russia from the Republic of Niger. Transportation was carried out in special containers with forced aeration, water temperature 23 ° С. !!… See (Table 1) was the experiment protocol also developed by you with due care? Double-check the steps.

Reply: 77-79 The sampling protocol, procedures employed and the experiment protocol were ethically reviewed and approved by the P.G. Demidov Yaroslavl State University (protocol 2019_10.01).

It is known that the water temperature in the reservoirs of the habitats of Polypterus senegalus varies from 15 to 40 ºÐ¡. (Offem et al. 2011, Eneogwe et al. 2022). Therefore, the temperature regime and operation mode chosen by the carrier are optimal.

Eneogwe C., Sanni I. M., Abubakar A. U., Abraham I. A. Seasonal variation of reservoir water quality: A case study of Kubanni reservoir, Zaria, Nigeria //Environmental Health Engineering And Management Journal. – 2022. – Т. 9. – â„–. 2. – С. 125-134.

Offem B. O., Ayotunde E. O., Ikpi G. U., Ochang S. N., Ada F. B. Influence of seasons on water quality, abundance of fish and plankton species of Ikwori Lake, South-Eastern Nigeria. Fisheries and Aquaculture Journal. 2011. V.13. P. 1-18. https://doi.org/10.4172/2150-3508.1000013

Comment 3: 3. Results:

The authors should be reduce this part, please the results shown in figures (2a-c).

Reply: 156-159 this description has been abbreviated.

Comment 4: 4. Discussion:

This part is very long, reduce it and discuss only the obtained results.

Reply: 350-386; 421-438 this description has been abbreviated.

Unfortunately, due to the lack of publications, we were unable to build similar series for such indicators as the number of capsules and their surface area. However, based only on the data on section diameters, it can be assumed that the filtration activity of the kidney of P. senegalus is comparable to that of freshwater teleosts and inferior to that of amphibians. ....

... there are other researches that assure this result ...

Comment 5: Our study showed that the ultrastructure of type I cells of the intermediate tubule 406 has a common structural plan with the epitheliocytes of P. dolloi.

Reply: 439-440 Our study showed that the ultrastructure of type I cells of the intermediate tubule has a common same structural with the epitheliocytes of P. dolloi

Comment 6: 5. Conclusion:

It would also be useful to include in the conclusions all the samples analyzed that showed different section diameters in order to assume that the filtration activity of the P. senegalus kidney is similar to other fish…Explain this step better.

Reply: Added a mention of filtration activity to the final conclusion.
